# The Association between 24 h Movement Guidelines and Internalising and Externalising Behaviour Problems among Chinese Preschool Children

**DOI:** 10.3390/children10071146

**Published:** 2023-06-30

**Authors:** Na Zhu, Hongzhi Guo, Dongmei Ma, Qiang Wang, Jiameng Ma, Hyunshik Kim

**Affiliations:** 1Department of Sports, Shenyang Ligong University, Shenyang 110159, China; yoyo70521@163.com; 2Graduate School of Human Sciences, Waseda University, Tokorozawa 3591192, Japan; hz-guo@moegi.waseda.jp; 3Department of Medicine and Science in Sports and Exercise, Graduate School of Medicine, Tohoku University, Sendai 9808576, Japan; dn-ma@sendai-u.ac.jp; 4College of Sports Science, Shenyang Normal University, Shenyang 110034, China; 13804999441@163.com; 5Faculty of Sports Sciences, Sendai University, Shibata 9891693, Japan; jm-ma@sendai-u.ac.jp

**Keywords:** preschool children, 24 h movement guidelines, internalising and externalising behavioural problems, mental health, Chinese

## Abstract

This study examined the relationship between adherence to 24 h movement guidelines (24 h MGs) and internalising and externalising behavioural problems in Chinese children aged 3–6 years, with a specific focus on the differences between weekdays and weekends. The guidelines include recommendations for physical activity (PA), screen time (ST), and sleep duration (SD). The results indicated a stronger association between adherence to these guidelines and behavioural problems on weekends compared to weekdays. Specifically, the odds of experiencing internalising problems were 1.33 higher (95% CI: 1.05–1.69) when not satisfying all three behaviours compared to not satisfying one or two. Moreover, on weekends, when ST was not fulfilled, there was a higher likelihood of externalising behaviour problems compared to when it was fulfilled (OR, 1.18, 95% CI, 1.01–1.38), and when all three behaviours were not met, the likelihood was even higher (OR, 1.50, 95% CI, 1.04–2.18). Children who met all three guidelines had fewer internalising and externalising behavioural problems, suggesting a potential beneficial effect on mental health. The study revealed that a higher adherence to these recommendations corresponded to a lower risk of mental health problems. Additionally, higher screen time was linked to an increase in externalising behavioural issues. These findings underscore the importance of adherence to 24 h MGs for optimal mental health in children. Future interventions should consider these behavioural factors and incorporate strategies to promote adherence to these guidelines, particularly on weekends.

## 1. Introduction

The World Health Organization (WHO) reports growing concerns regarding the scope and impact of mental illness in children, with an estimated 10–20% of children and adolescents worldwide suffering from mental disorders such as depression, anxiety, and aggressive behaviours [1]. Approximately 70% of mental illnesses develop during childhood and adolescence, and it is predicted that 15–25% of the world population will experience at least one mental health problem before the age of 19 years [2]. In China, the occurrence of mental disorders among children aged 6–16 years is a significant health concern, with a prevalence rate ranging from 9.49% to 15.24% [3]. Considering that mental health problems are likely to begin in early childhood [4] and continue into adulthood [5], the early identification of risk factors for children’s mental health is crucial.

The strengths and difficulties questionnaire (SDQ) is an appropriate tool for screening behavioural and psychopathology issues in children aged 3–16 years, as it can detect internalising and externalising behavioural problems in children [6]. Externalising behaviours are a range of aggressive and uncontrolled behaviours such as aggression, conduct problems, hostility, hyperactivity, and attention issues. Internalising behaviours are actions directed towards oneself and can manifest as anxiety, fear, sadness, depression, social withdrawal, and physical dissatisfaction [7,8]. These two behavioural problems co-occur at high rates among children and are likely to lead to mental disorders in adulthood [9]. Previous studies have reported that irregular PA [10], excessive ST [11], and short SD are associated with internalising and externalising behaviour problems [12]. Nevertheless, prior research has examined these behaviours in isolation, and has disregarded their interconnectedness [13,14]. Our hypothesis posits that achieving the recommended 24 h Movement Guidelines (24 h MGs) for physical activity (PA) in Chinese preschool children may be associated with a decrease in both internalising and externalising behaviour problems.

The interdependence of PA, screen time (ST), and sleep duration (SD) has been established by previous studies [15,16]. Moreover, there exists compelling evidence that supports the notion of an interaction between the health benefits of movement behaviours within a 24 h time frame [17,18,19]. Consequently, in light of the publication of the 24 h MGs for Canadian youth and adolescents (aged 5–17 years) [20], there is a necessity to comprehend the proportion of children who adhere to the guidelines, and to determine which specific amalgamation of behaviours holds the greatest significance in relation to their mental wellbeing. Furthermore, kindergarten children typically experience a less structured and more autonomous environment on weekends, whereas structured and detailed schedules characterize their weekdays. A recent meta-analysis reported that the 24 h MGs of adolescents are generally healthier on days with more structure [21], although there have been no studies, to the author’s knowledge, investigating the impact of structure on children’s behaviour. Therefore, this study aimed to investigate the potential link between adherence to 24 h MGs and internalising and externalising behavioural problems in Chinese children aged 3–6 years, as well as any differences in adherence between weekdays and weekends. In doing so, this study seeks to contribute to promoting mental health in Chinese children and developing evidence-based interventions to address any potential issues that may arise. We hypothesised that meeting 24 h MGs in Chinese preschool children would contribute to a reduction in internalising and externalising behaviour problems.

## 2. Materials and Methods

### 2.1. Study Design and Participants

This cross-sectional study partially extracted data from the “International Joint Study for the Improvement of 24 h Movement and Mental Health of East Asian Children”. Accelerometers were used to measure PA and sedentary behaviour from September to October 2022, and a questionnaire survey was administered to investigate internalising and externalising behavioural problems. Regarding participant recruitment, kindergartens in Shenyang, China, were selected through convenience sampling. Per the Declaration of Helsinki, informed consent was obtained from the parents and teachers of children aged 3–6 years from the selected kindergartens in Shenyang, China. Data from 200 children (51.0% boys, 49.0% girls) who provided signed consent forms were collected and analysed. The participation recall rate was 74.6%. The study received prior approval from the Sendai University Ethics Committee, Faculty of Sports Science (SU2021-05).

### 2.2. Measurements

#### 2.2.1. Physical Activity

The study used a triaxial accelerometer (Active Style Pro HJA-750C; Omron Health Care Co., Ltd., Kyoto, Japan) to quantify PA. The accelerometer is capable of generating the metabolic equivalent of task (MET) values by utilizing predictive equations specifically designed for adults [22]. However, it is important to note that MET values can overestimate results for children, as previously reported [23]. To account for the overestimation of MET values in children compared to adults when using the Active Style Pro accelerometer, the researchers used conversion equations obtained from previous studies.

The study protocol required the participants to wear a triaxial accelerometer around their waist for one week, starting from the time they woke up until their bedtime, which was set between 7:00 and 21:00, except during showering or swimming. If the accelerometer recorded a value of 0 for a duration of 20 min or longer, it was inferred that the individual was not using the wearable device. The accelerometer was used to quantify the duration of sedentary behaviour (defined as activities with a metabolic equivalent of 1.5 or less), light-intensity PA (defined as activities with a metabolic equivalent ranging from 1.6 to 2.9), and moderate to vigorous PA (defined as activities with a metabolic equivalent of 3 or greater). The data were collected at 10-s intervals. To determine the daily PA levels, only the data from days when participants wore the accelerometer for 600 min or more were considered. Additionally, the study required a minimum of four days per week for data inclusion [24]. We calculated the children’s weekly average MVPA (weekly average MVPA = (weekday MVPA) + (weekend MVPA)/2) [25,26].

#### 2.2.2. Screen Time

ST refers to time spent on screen-based behaviours, including recreational, stationary, sedentary, and active screen time [27]. ST data were collected by asking parents about their child’s screen-based behaviours in the past week. Two questions were used to assess screen time. First, “What is the average amount of time that your offspring spends viewing television or DVDs on a daily basis”? The second question pertains to the average duration a child spends using electronic gadgets, such as smartphones and tablets, on a daily basis. The respondents’ answers were obtained in written form, pertaining to the average duration of screen time that their children engage in on both weekdays and weekends. In order to determine the mean duration of screen-viewing activities per week, the researchers multiplied the mid-category values of daily activity duration by the number of days the child engaged in screen-viewing activities during both weekdays and weekends. The researchers proceeded to compute the mean daily screen time of the children by applying the following formula: average daily ST = ((weekday ST × 5) + (weekend ST × 2)/7) [28,29].

#### 2.2.3. Sleep Duration

SD was evaluated through parental inquiry, specifically by asking the following question: “What is the duration of your child’s nocturnal sleep”? The study computed the average daily SD for weekdays and weekends by applying the following formula: average daily SD = ((weekday SD × 5) + (weekend SD × 2)/7) [30].

#### 2.2.4. Strengths and Difficulties Questionnaire

The SDQ was employed to assess the internalising and externalising behaviours of children [31]. The SDQ is a standardized tool comprising 25 items that assess diverse dimensions of a child’s mental health and behaviour. The survey comprises five distinct subscales, namely, emotional symptoms, conduct problems, hyperactivity/inattention, peer relationship problems, and prosocial behaviour. Each subscale consists of five items that are evaluated using a Likert scale. The aforementioned subscales can be divided into two distinct scales, namely, internalising behaviours (including emotional symptoms and peer relationship problems) and externalising behaviours (including conduct problems and hyperactivity/inattention) [32]. The Chinese version of the SDQ has been demonstrated to be reliable in previous research [33].

#### 2.2.5. Adherence to the 24 h Movement Guidelines

Adherence to the 24 h MGs for children was assessed using three recommendations [26,34]: adherence to PA guidelines (consisting of 180 min of total PA, with at least 60 min of moderate to vigorous PA per day), compliance with ST guidelines (limited to less than one hour per day), and adherence to SD guidelines (ranging from 10 to 13 h within a 24 h period).

#### 2.2.6. Demographic Variables

A survey questionnaire was employed to collect demographic data from the study participants. The questionnaire included items on the children’s sex, age, bedtime, wake-up time, weight, height, sports club participation, as well as their mother’s employment status, education level, number of media devices, number of family members, height, and weight. The parents of the children provided this information. The researchers measured the children’s height and weight using units of 0.1 cm and 0.1 kg, respectively. BMI z-scores were computed using the World Health Organization’s growth criteria [35]. Additionally, fat mass was computed using the formulas provided by Wang et al. [36].

### 2.3. Statistical Analysis

In this study, data from 200 young Chinese children (102 boys and 98 girls) who provided complete information on the study variables were analysed using three models. The initial model examined the differences in demographic variables among the children and guardians, encompassing age, stature, mass, body mass index, and adipose tissue. Statistical analyses in this study involved independent t-tests for continuous variables and chi-squared tests for categorical variables. The second model compared differences in children’s PA, ST, and SD between weekdays and weekends, using independent *t*-tests to examine adherence to the 24 h MGs (i.e., PA, ST, and SD) between weekday and weekend samples. The third model employed a logistic regression analysis to investigate the correlation between compliance with individual 24 h MGs variables, and internalising and externalising behaviours during weekdays and weekends. Age, BMI, and fat mass were used as covariates. Ternary plots were used to display the data distribution, visualizing the relationships among MVPA, ST, and SD, while also displaying a colour-coded distribution of SDQ scores for internalising or externalising problems. Ternary plots are an efficient technique for illustrating the relationships between three variables, the proportions of which sum to a constant (100%). They can be viewed as scatterplots of compositions that present all possible combinations of variables in a single graph, making it easier to identify underlying patterns or trends [16]. Analyses and visualizations were conducted using R version 4.2.3. For statistical significance, *p*-values were set at <0.05. Data were analysed using SPSS version 26.0 (IBM, Armonk, NY, USA).

## 3. Results

### 3.1. Sample Characteristics

Table 1 compares the children’s and parents’ characteristics between sex. Analysis of the characteristics of the children revealed that boys tended to have higher BMI and fat mass values than girls (*p* < 0.049, *p* < 0.003, respectively). Regarding participation in sports clubs, data analysis showed that girls had a higher participation rate (63.3%) than boys (47.1%). Parental characteristics did not show a significant relationship, but overall, the full-time employment rate and the rate of parents with college or higher education were both high.

### 3.2. Comparison of PA, Sedentary Behaviours, and Sleep on Weekdays and Weekends

The study found that among the PA measures, the MVPA and steps taken were higher on weekends compared to weekdays (MVPA: 82.4 ± 20.5 min vs. 72.1 ± 20.9 min on weekends and weekdays, respectively (*p* < 0.001) (steps: 9822.2 ± 3416.0 vs. 9210.3 ± 2062.6 on weekends and weekdays, respectively (*p* < 0.001)). Conversely, light PA showed higher results on weekdays compared to weekends (318.7 ± 44.8 min vs. 302.7 ± 44.4 min, *p* < 0.031). Among the sedentary behaviour items, TV/DVD and smartphone/tablet use were higher on weekends than weekdays (TV/DVD: 32.9 ± 35.7 min vs. 53.9 ± 57.8 min on weekdays and weekends, respectively (*p* < 0.001); smartphone/tablet: 20.8 ± 27.2 min vs. 31.7 ± 37.2 min on weekdays and weekends, respectively (*p* < 0.001)). Among the sleep items, bedtime, wake-up time, and sleep time were found to be later on weekends than weekdays (bedtime: 448.8 ± 39.0 min vs. 415.4 ± 22.2 min on weekends and weekdays, respectively (*p* < 0.001); wake-up time: 1310.9 ± 38.5 min vs. 1296.5 ± 35.2 min on weekends and weekdays, respectively; sleep duration: 578.0 ± 41.6 min vs. 559.0 ± 35.3 min on weekends and weekdays, respectively (*p* < 0.001; Table 2).

### 3.3. Ternary Plot of the Composition of 24 h MGs of SDQ Scores for Internalising/Externalising Behaviour Problems

Using ternary plots, Figure 1 shows the average distribution of children’s MVPA, ST, and SD throughout the week. A colour-coded distribution of the SDQ scores for internalising and externalising problems is also shown. A yellow tendency indicates a lean toward normalcy, whereas a purple indicator denotes an inclination toward the presence of issues. Moreover, it can be observed that as the proportion of screen time (ST) increases, there is a greater tendency for externalising problems.

### 3.4. Association between 24 h Movement Behaviours and Internalising/Externalising Problems

Table 3 shows a multivariate analysis of the adjusted ORs of 24 h MG constructs on weekdays and weekends for children’s internalising and externalising behavioural problems. Significant differences were observed only on weekends for internalising behavioural problems. The odds of experiencing these problems were 1.33 higher (95% CI:1.05–1.69) when not satisfying the guideline requirements for all three behaviours (PA, ST, and SD) compared to not satisfying one or two. Moreover, the odds of having behavioural problems were higher when combining PA + SD (OR, 1.49, 95% CI, 1.11–1.98), ST + PA (OR, 1.31, 95% CI, 1.00–1.70), or having none of the three behaviours than when meeting all behaviour recommendations (OR, 1.70, 95% CI, 1.08–2.69). Regarding externalising behavioural problems, the study found that on weekdays, when all three behaviour recommendations (PA, ST, and SD) were not met, the likelihood of such problems was lower (OR: 0.75, 95% CI, 0.61–0.92). However, on weekends, when sleep time was not fulfilled, there was a higher likelihood of such problems (OR, 1.18, 95% CI, 1.01–1.38), and when all three behaviour recommendations were not met, the likelihood was even higher (OR, 1.50, 95% CI, 1.04–2.18; Table 3).

## 4. Discussion

This study hypothesis was that adherence to 24 h MGs would be associated with reduced internalising and externalising behavioural problems in Chinese preschool children aged 3–6 years. To test this hypothesis, we analysed the relationship between adherence to 24 h MGs and internalising and externalising behavioural problems. We further examined whether there were differences in adherence between weekdays and weekends.

First, the results for internalisation confirmed that the likelihood of experiencing internalising behaviour problems increased in the following order: only meeting the ST and PA recommendations (ST + PA), only meeting the PA and SD recommendations (PA + SD), and not meeting any of the three recommendations (none), as compared to meeting all three recommendations. Overall, our results indicate that meeting all three recommendations (PA, ST, and SD) was associated with improved mental health in children compared to a combination of only meeting ST and SD recommendations. This finding is consistent with previous research [12,37,38]. However, our study adds new insights into the role of weekends in the association between 24 h MG adherence and behavioural problems. This study’s results show that the higher the number of recommendations met (PA, ST, and SD), the lower the risk of internalisation and externalisation problems. These findings suggest that there are mental health benefits associated with adhering to 24 h MGs. Numerous studies have reported that internalised behavioural problems, such as anxiety, fear, sadness, depression, social withdrawal, and physical dissatisfaction, are associated with a high risk of persistently exhibiting internalised behavioural problems from early childhood to adolescence [39,40]. Furthermore, children with internalising behavioural problems often face challenges forming positive peer relationships, and are more prone to isolating behaviours and social withdrawal [41].

To address and improve children’s internalised behavioural problems, exploring their relationship with PA, ST, and SD is crucial. Numerous studies have demonstrated that regular engagement in PA is associated with positive mental health outcomes. These studies suggest that PA acts as a protective factor against internalised behavioural problems, including symptoms of depression and anxiety, regardless of age and geographical location [42,43]. Furthermore, it has been found that increased sedentary time, particularly ST, is associated with anxiety, emotional issues, and peer problems [44]. Excessive ST can also decrease the time spent on PA, thus limiting the potential benefits of PA on mental health [45]. While some studies have demonstrated an independent correlation between high ST and these difficulties, recent research suggests that a synergistic effect can be achieved by combining recommended levels of PA with low ST to reduce the risk of emotional difficulties in children [46]. Previous studies have reported that a shorter SD negatively affects the levels of neurotransmitters involved in regulating mood and cognition [47], and experiencing sleepiness and fatigue during morning activities can make it more difficult to maintain a healthy and active lifestyle [48,49]. However, our study found that SD was longer on weekends than on weekdays, despite bedtimes and wake-up times being later on weekends. Previous research has also reported that children with later bedtimes experience symptoms such as anxiety and depression [50], and that irregular sleeping–waking patterns can significantly affect their adaptability in kindergarten settings [51]. Although many studies have reported an association between sleep duration satisfaction and health-related factors, further studies are needed to examine the effects of late bedtime, late awakening, and poor sleep quality.

In addition, our study’s results indicate a higher likelihood of externalising behavioural problems when ST exceeds the recommended amount. Our findings align with studies conducted in other countries using the SDQ to measure the socio-emotional development of preschool children, which have reported an association between increased ST and externalising behavioural problems such as hyperactivity, emotional symptoms, conduct issues, and poor prosocial behaviour [52,53,54]. With the advancement of media technology and increased accessibility, high ST among children in Asia has been highlighted as a significant concern [55]. Our study’s results revealed a favourable association between externalising behaviour problems in children when all three recommended activity guidelines were met. This finding is consistent with previous research that has shown an association between conduct issues, hyperactivity, and meeting these guidelines [56], and other studies have reported better mental health outcomes for individuals who fulfil a greater number of PA behaviour recommendations [18]. Externalising behaviours, such as aggression and destructive behaviour, tend to decrease with the development of cognitive abilities and emotion regulation [57]. However, a small percentage (5–7%) of antisocial children may exhibit persistent or worsening externalising problems [58]. Furthermore, both internalising and externalising behaviours in children can negatively affect children, their families, and society in both the short and long term [59,60]. Therefore, based on our research findings, it is important to identify risk factors for mental health problems early to mitigate potential negative impacts. According to a meta-analysis, children’s movement behaviours are reported to be healthier on weekdays than on weekends, as weekdays are characterised by less autonomy, more structure, and less fragmented routines under adult supervision [21]. Further, children tend to sleep more on weekends than weekdays [61]. As a result, wakeful activities (i.e., sedentary behaviour and PA) decreased on weekends. However, for children, greater autonomy during weekends may lead to a preference for sedentary behaviour, particularly recreational screen time, over PA. Therefore, future intervention studies should consider an ecological model [62] including the individual, family, and environmental factors to examine 24 h behaviours during weekends and ensure that the three recommended guidelines are fulfilled.

Several limitations should be considered when interpreting our findings. First, the cross-sectional design of this study does not allow for the confirmation of a causal relationship between the composition of the 24 h MGs and mental health in Chinese children aged 3–6 years. A longitudinal or interventional study is necessary to establish a clear relationship between these variables. Second, given that our study focused on preschoolers aged 3–6 years in northeast China, concerns arise regarding the extent to which the findings can be generalised to all young children in China. Third, similar to other studies involving preschool children, our study utilised a combination of objective measurements such as accelerometers and questionnaire surveys completed by parents or family members who observed the child’s behaviour. It is important to acknowledge that the parents’ psychological state can impact not only their evaluation of their children’s behaviour but also their responses to the study [63]. Therefore, the results should be interpreted with caution. The results of this study can strengthen the evidence for various factors related to mental health in preschool children, specifically concerning the combination of 24 h MGs. As this study was limited to preschool children in Asian countries, it highlights the need to develop effective intervention programs in the future. These findings hold significant importance in guiding future interventions to promote preschool children’s health and wellbeing.

## 5. Conclusions

This study investigated the relationship between Chinese preschool children’s compliance with 24 h MGs and internalising and externalising behavioural problems. The results showed that on weekends, children who did not meet all three movement guidelines were more likely to exhibit internalised behavioural problems. Children who did not engage in a specific combination of behaviours, such as PA and ST, or those who did not engage in any of the three behaviours were also more likely to exhibit internalised behavioural problems. Our findings will help inform future updates to the 24 h MGs for early childhood to ensure optimal movement guidelines and comprehensive mental health development in young children.

## Figures and Tables

**Figure 1 children-10-01146-f001:**
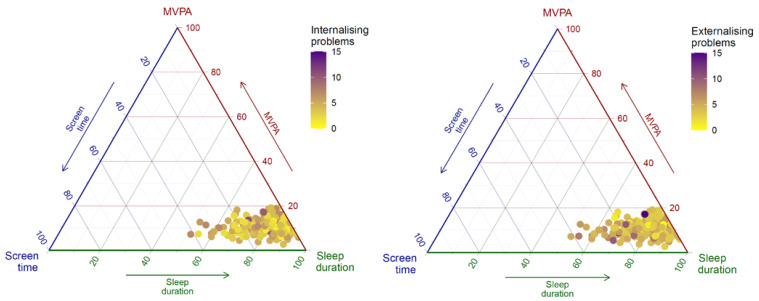
The ternary plot shows the composition of 24 h MGs and SDQ scores for internalising/externalising behaviour problems.

**Table 1 children-10-01146-t001:** Characteristics of the participants.

	Total (*n* = 200)	Boys (*n* = 102)	Girls (*n* = 98)	*p*-Value
Children’s				
Age (months: mean, SD)	57.5 ± 10.0	56.9 ± 10.4	58.1 ± 9.5	0.428
Height (cm: mean, SD)	110.0 ± 8.0	109.9 ± 8.7	110.1 ± 7.2	0.901
Weight (kg: mean, SD)	19.7 ± 4.1	20.0 ± 4.0	19.4 ± 4.3	0.311
BMI (kg/m^2^: mean, SD) ^a^	16.2 ± 2.0	16.4 ± 1.9	15.9 ± 2.1	0.049
Z-score (mean, SD)	0.3 ± 1.2	0.5 ± 1.3	0.2 ± 1.1	0.064
FM (mean, SD) ^b^	3.1 ± 2.1	3.5 ± 2.0	2.6 ± 2.1	0.003
Sports club				
Yes	110 (55.0)	48 (47.1)	62 (63.3)	0.021
No	90 (45.0)	54 (52.9)	36 (36.7)	
SDQ ^c^				
Internalising problems: IP	3.9 ± 1.9	3.9 ± 1.9	3.8 ± 1.9	0.891
Externalising problems: EP	4.2 ± 2.0	4.1 ± 2.1	4.4 ± 1.8	0.067
Total Difficulties	8.1 ± 2.9	7.9 ± 3.1	8.3 ± 2.7	0.250
Parents’ (Mother’s)				
Height (cm: mean, SD)	160.0 ± 5.0	159.1 ± 5.1	160.3 ± 4.9	0.014
Weight (kg: mean, SD)	55.1 ± 6.8	54.7 ± 6.9	54.8 ± 6.5	0.904
BMI (kg/m^2^: mean, SD)	21.5 ± 2.3	21.6 ± 2.4	21.3 ± 2.2	0.160
Number of family members	3.6 ± 1.1	3.6 ± 1.1	3.7 ± 1.2	0.921
Number of media	6.5 ± 2.4	6.5 ± 2.4	6.6 ± 2.4	0.748
Employment status				
Full time	150 (75.0)	77 (75.5)	73 (74.5)	0.809
Part time	14 (7.0)	6 (5.9)	8 (8.2)	
Homemaker	36 (18.0)	19 (18.6)	17 (17.3)	
Education level				
High school or lower	4 (2.0)	2 (2.0)	2 (2.0)	0.882
Tech school or college	60 (30.0)	29 (28.4)	31 (31.6)	
University or above	136 (68.0)	71 (69.6)	65 (66.4)	

Abbreviations: SD, standard deviation; FM, fat mass; BMI, body mass index. ^a^ BMI percentiles of the participants were classified according to the age criteria of the Center for Disease Control and Prevention growth charts for the United States [34]. ^b^ FM was calculated using the height–weight equation [25]. The statistical significance of the continuous variables was determined using a *t*-test, while the categorical variables were analysed using a chi-square test. ^c^ SDQ, strengths and difficulties questionnaire. externalising problems, conduct problems + hyperactivity/inattention problems; internalising problems, emotional symptoms + peer relationship problems [32].

**Table 2 children-10-01146-t002:** Comparison of PA, sedentary behaviours, and sleep on weekdays and weekends.

	Weekdays	Weekends	*p*-Value
	Mean ± SD		
Physical activity (min/day)			
MVPA	72.1 ± 20.9	82.4 ± 20.5	<0.001
LPA	318.7 ± 44.8	302.7 ± 44.4	<0.001
Steps	9210.3 ± 2062.6	9822.2 ± 3416.0	0.031
Wear time (min/day)	652.3 ± 69.3	652.8 ± 73.5	0.945
Sedentary behaviour (min/day)			
Sedentary time	261.5 ± 55.3	267.8 ± 51.5	0.243
TV/DVD	32.9 ± 35.7	53.9 ± 57.8	<0.001
Smartphone/tablet	20.8 ± 27.2	31.7 ± 37.2	0.001
Sleep (min/day)			
Wake-up time	415.4 ± 22.2	448.8 ± 39.0	<0.001
Bedtime	1296.5 ± 35.2	1310.9 ± 38.5	<0.001
Sleep duration	559.0 ± 35.3	578.0 ± 41.6	<0.001

Abbreviations: SD, standard deviation; MVPA, moderate to vigorous physical activity; LPA, light physical activity. A *t*-test was used to calculate *p*-values for continuous variables.

**Table 3 children-10-01146-t003:** Associations between meeting the combinations of the 24 h MGs and internalising/externalising problems among Chinese young children.

	Internalising Problems		Externalising Problems	
	Weekday	Weekend	Weekday	Weekend
	OR (95% CI)	OR (95% CI)	OR (95% CI)	OR (95% CI)
Model 1				
PA				
Meet	1	1	1	1
Do not meet	1.14 (0.97, 1.35)	1.03 (0.82, 1.29)	1.15 (0.99, 1.35)	1.06 (0.86, 1.30)
ST				
Meet	1	1	1	1
Do not meet	1.06 (0.89, 1.26)	1.10 (0.94, 1.28)	1.05 (0.90, 1.24)	1.18 (1.01, 1.38)
SD				
Meet	1	1	1	1
Do not meet	1.12 (0.90, 1.38)	1.05 (0.89, 1.22)	1.14 (0.92, 1.42)	0.99 (0.85, 1.14)
Model 2				
All Meet	1	1	1	1
PA + SD	1.19 (0.69, 2.06)	1.49 (1.11, 1.98)	1.03 (0.60, 1.75)	1.27 (0.97, 1.65)
ST + SD	1.52 (0.94, 2.44)	1.45 (0.87, 2.44)	0.84 (0.50, 1.38)	1.02 (0.60, 1.73)
ST + PA	1.30 (0.92, 1.83)	1.31 (1.00, 1.70)	1.03 (0.75, 1.40)	1.11 (0.86, 1.42)
None	1.29 (0.86, 1.94)	1.70 (1.08, 2.69)	1.40 (0.99, 1.98)	1.50 (1.04, 2.18)

Abbreviations: PA, physical activity; ST, screen time; SD, sleep duration. OR (95% CI): Odds ratio (95% confidence intervals). Adjusted analyses included children’s age, sex, and BMI. Meeting the recommendations is defined as 180 min of total PA including more than 60 min/day of moderate-to-vigorous PA, no more than 60 min/day for screen time, and between 10 and 13 h/day for sleep duration.

## Data Availability

The data are included in the article.

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
