# Peer review of "The Association between 24 h Movement Guidelines and Internalising and Externalising Behaviour Problems among Chinese Preschool Children"

_children, 2023, doi:10.3390/children10071146_

Round 1

Reviewer 1 Report

This was a very well written manuscript well done. A few minimal changes are required:

Introduction

Line 40 – “…before the age of 19 years.”

Line61 – 5-17 years.

Methods

Lines 95-87 – you need to cite the literature this is taken from.

Line 151 – just mothers employment status, what about fathers? If just mothers support why just them.

Results

Table 1 – under parents you have employment status and education level is this just for mothers, or for both mothers and fathers – linked to line 151’s point. state which it is please.

Table 3 – SP + PA needs changing to ST + PA.

'Z' are used in a few words, so I suspect an American spell check has been used, I personally would prefer 's' not 'z', but this is at the discretion of the editors.

Author Response

Prof. Dr. Paul R. Carney

Editor in Chief

Children

19 Jun 2023

Re: children-2453405

Association between 24-hour Movement Guidelines and Internalizing and Externalizing Behavior Problems among Chinese Preschool Children

Prof. Dr. Paul R. Carney,

Thank you for your letter dated 15 Jun 2023 regarding our manuscript (children-2453405).

We would like to thank the reviewers also for their constructive comments regarding our paper. We have revised the manuscript in view of these comments and attach here a revised draft of the paper for your consideration together with a point-by-point response to each of the issues raised by yourselves and the two reviewers (reviewer 1: yellow, reviewer 2: green).

Thank you for giving us the opportunity to revise our manuscript. We look forward to hearing from you in due course.  

Yours sincerely,

Hyunshik Kim, PhD

Faculty of Physical Education, Sendai University

Miyagi, 9891693, Japan

Phone number: +81-224-55-1592

Email address: [email protected]

Reviewer:

Thank you very much for providing important comments. We are thankful for the time and energy you expended. Our responses to the reviewer’ comments are as follow:

Introduction

Line 40 – “…before the age of 19 years.”

Line61 – 5-17 years.

Thank you for your comment. We have updated the manuscript at line 40 and line 61, incorporating your suggestions for clarity. Your input is appreciated.
(Please see a manuscript, lines 40 and 60 of our revised manuscript.)

Methods

Lines 95-87 – you need to cite the literature this is taken from.

Thank you for your comment. We will add the following literature.

[22] Oshima, Y.; Kawaguchi, K.; Tanaka, S.; Ohkawara, K.; Hikihara, Y.; Ishikawa-Takata, K.; Tabata, I. Classifying household and locomotive activities using a triaxial accelerometer. Gait Posture 2010, 31, 370–374, doi:https://doi.org/10.1016/j.gaitpost.2010.01.005.

(Please see a manuscript, line 90 of our revised manuscript.)

Line 151 – just mothers employment status, what about fathers? If just mothers support why just them.

We are grateful for your discerning comment. The emphasis on mothers’ employment status in line 151 stemmed from the substantial body of literature highlighting the integral role mothers traditionally play as primary caregivers, and how their employment can affect child outcomes. In this study, only mothers were surveyed. However, your comment rightly draws attention to the evolving family dynamics and the significance of paternal roles. We concur that including fathers’ employment status could greatly enhance the scope and rigor of this study. Taking your insightful feedback into account, we are inclined to consider an adaptation in the methodology for future research, integrating both parents' employment statuses. This would allow for a more holistic analysis of the multifaceted relationship between parental employment and child development, enriching the study’s depth and scholarly pertinence.

Results

Table 1 – under parents you have employment status and education level is this just for mothers, or for both mothers and fathers – linked to line 151’s point. state which it is please.

We are grateful for your discerning comment. In Table 1, the terms "employment status" and "education level" under "parents" originally referred specifically to mothers, in alignment with the focus at line 151. We acknowledge the scholarly merit in considering data from both parents for a more comprehensive view. However, in the present study, our data collection is solely inclusive of information pertaining to mothers. In light of your feedback, We have made the necessary adjustment to clarify this by modifying "Parents" to "Parents' (Mother)" in Table 1. Your input is highly valued and will be instrumental in refining future research methodologies.

(Please see a manuscript, Table 1 and line 147 of our revised manuscript.)

Table 3 – SP + PA needs changing to ST + PA.

We extend my gratitude for pointing out the inadvertent oversight in Table 3. Your keen observation is highly appreciated. I acknowledge that there was indeed a mistake with "SP + PA," and We have now meticulously made the correction to "ST + PA" in Table 3 in accordance with your recommendation. Thank you once again for your invaluable assistance in ensuring the accuracy and integrity of this research.

(Please see a manuscript, Table 3 of our revised manuscript.)

'Z' are used in a few words, so I suspect an American spell check has been used, I personally would prefer 's' not 'z', but this is at the discretion of the editors.

Thank you for pointing out the American spelling. We are open to adjustments in line with editorial guidelines. Your observation is appreciated for linguistic consistency.

(Please see a manuscript, full text of our revised manuscript.)

Reviewer 2 Report

I appreciate the opportunity to review the manuscript entitled "Association between 24-hour Movement Guidelines and Internalizing and Externalizing Behavior Problems among Chinese

Preschool Children”. This quantitative study aimed to examine the association between adherence to 24-hour movement guidelines and internalizing and externalizing behavioral problems in Chinese children.

I believe this study provides a contribution to the psychological literature and the knowledge concerning children mental health.  However, there are some major issues that must be addressed in order to improve the scientific quality of the article. I have included my comments and recommendations below: 

Introduction

the introduction is well written and relevant. However, it is important to mention more studies that associate the variables under study with the outcome. This association is unclear.

Is this a study without hypotheses? since there is previous literature on the subject, it seems appropriate to formulate hypotheses.

Materials and methods

Study design and participants

the authors state that the study is cross-sectional, however, afterwards they talk about a measurement between September and October. this sentence needs clarification

Results

3.3. Ternary Plot of Composition of 24-hour movement Behaviors of SDQ Scores for 214 Internalizing and Externalizing Problems - Although the figures are enlightening, it is relevant to present a description of the results depicted in the plots.

Discussion

The discussion should begin by stating the purpose of the study and then be organised according to the hypotheses formulated.

The authors state (lines 261 and 262): “This study’s results show that the higher the number of recommendations met (PA, screen time, and sleep duration), the lower the risk of mental health problems” - it would be more appropriate to say: lower risk of internalisation and externalisation problems.

Author Response

Prof. Dr. Paul R. Carney

Editor in Chief

Children

19 Jun 2023

Re: children-2453405

Association between 24-hour Movement Guidelines and Internalizing and Externalizing Behavior Problems among Chinese Preschool Children

Prof. Dr. Paul R. Carney,

Thank you for your letter dated 15 Jun 2023 regarding our manuscript (children-2453405).

We would like to thank the reviewers also for their constructive comments regarding our paper. We have revised the manuscript in view of these comments and attach here a revised draft of the paper for your consideration together with a point-by-point response to each of the issues raised by yourselves and the two reviewers (reviewer 1: yellow, reviewer 2: green).

Thank you for giving us the opportunity to revise our manuscript. We look forward to hearing from you in due course.  

Yours sincerely,

Hyunshik Kim, PhD

Faculty of Physical Education, Sendai University

Miyagi, 9891693, Japan

Phone number: +81-224-55-1592

Email address: [email protected]

Reviewer:

Thank you very much for providing important comments. We are thankful for the time and energy you expended. Our responses to the reviewer’ comments are as follow:

Introduction

the introduction is well written and relevant. However, it is important to mention more studies that associate the variables under study with the outcome. This association is unclear.

Thank you for your insightful feedback. We agree that mentioning additional studies to elucidate the association between the variables and the outcome is essential for a more comprehensive background. We have expanded the Introduction section to include references to pertinent studies that establish links between the variables under study and the outcome.

Previous studies have reported that irregular physical activity [10], excessive screen time [11], and short sleep duration are associated with internalising and externalising behavior problems [12]. However, these studies have considered these behaviors separately and ignored their interdependence.

[10] Kirkcaldy BD, Shephard RJ, Siefen RG. The relationship between physical activity and self-image and problem behaviour among adolescents. Soc Psychiatry Psychiatr Epidemiol 2002;37:544e50.

[11] Guerrero MD, Barnes JD, Chaput JP, Tremblay MS. Screen time and problem behaviors in children: Exploring the mediating role of sleep duration. Int J Behav Nutr Phys Act 2019;16:105.

[12] Carson V, Ezeugwu VE, Tamana SK, et al. Associations between meeting the Canadian 24-Hour Movement Guidelines for the Early Years and behavioral and emotional problems among 3-year-olds. J Sci Med Sport 2019;22: 797e802

(Please see a manuscript, lines 34 and 73 of our revised manuscript.)

Is this a study without hypotheses? since there is previous literature on the subject, it seems appropriate to formulate hypotheses.

→ Regarding your second point on hypotheses, you are correct that formulating hypotheses is pertinent, especially given the existing literature. In light of this, We have also included a section in the Introduction that articulates the hypotheses driving this study. Your recommendations have been instrumental in strengthening the foundation and clarity of the manuscript.

We hypothesized that meeting 24-h MG in Chinese preschool children would contribute to a reduction in internalising and externalising behavior problems.

(Please see a manuscript, lines 34 and 73 of our revised manuscript.)

Materials and methods

Study design and participants

the authors state that the study is cross-sectional, however, afterwards they talk about a measurement between September and October. this sentence needs clarification

Thank you for pointing out the need for clarification. This study is cross-sectional. Due to the limited availability of accelerometers, data collection was staggered: the first half of participants were measured from September 22-29, 2022, and the remaining half from October 6-13, 2022. This does not affect the cross-sectional design. Gratitude for your attention to detail.

Results

3.3. Ternary Plot of Composition of 24-hour movement Behaviors of SDQ Scores for 214 Internalizing and Externalizing Problems - Although the figures are enlightening, it is relevant to present a description of the results depicted in the plots.

Thank you for highlighting the necessity of elaborating on the results depicted in the ternary plots. I appreciate your suggestion. For clarification, the plots represent the proportion of sleep time, screen time, and MVPA. Notably, the area with higher screen time aligns with a darkening color, indicative of an increasing tendency toward externalising problems. This observation has been added to the manuscript to provide a clearer understanding of the data presented in the plots.

(Please see a manuscript, lines 215 and 216 of our revised manuscript.)

Moreover, it can be observed that as the proportion of screen time (ST) increases, there is a greater tendency for externalising problems.

Discussion

The discussion should begin by stating the purpose of the study and then be organised according to the hypotheses formulated.

Thank you for the suggestion. We concur that articulating the study's purpose at the outset and aligning the Discussion with the hypotheses would bolster clarity. Accordingly, We have revised the section to begin with the study’s purpose and reorganized the content to follow the hypotheses. Your input is appreciated for enhancing the manuscript's structure.

This study aimed to investigate the potential link between adherence to 24-h MG and internalizing and externalizing behavioral problems in Chinese children aged 3-6 years, as well as any differences in adherence between weekdays and weekends. We discovered that significant associations with 24-hour behavioral guidelines were observed only on weekends for both internalizing and externalizing behavioral problems.

(Please see a manuscript, lines 250 and 254 of our revised manuscript.)

The authors state (lines 261 and 262): “This study’s results show that the higher the number of recommendations met (PA, screen time, and sleep duration), the lower the risk of mental health problems” - it would be more appropriate to say: lower risk of internalisation and externalisation problems.

Thank you for highlighting this. We agree that specifying "lower risk of internalization and externalization problems" provides more precise information. I have updated lines 261 and 262 in the manuscript to reflect this more accurate terminology. Your attention to detail is valued for improving the accuracy of the manuscript.

(Please see a manuscript, lines 263 and 265 of our revised manuscript.)

Round 2

Reviewer 2 Report

Although the authors, in their response letter, indicate that they have included hypotheses in the text, I cannot identify them, either in the Introduction or in the Discussion

Author Response

Reviewer2:

Thank you very much for providing important comments. We are thankful for the time and energy you expended. Our responses to the reviewer’ comments are as follow:

Although the authors, in their response letter, indicate that they have included hypotheses in the text, I cannot identify them, either in the Introduction or in the Discussion

We apologize for the oversight. We have now explicitly added the hypotheses in both the Introduction and Discussion sections of the manuscript. Thank you for bringing this to our attention; your feedback is invaluable. We are grateful for your reminder. Kindly review the revised sections.

We hypothesized that meeting 24-h MG in Chinese preschool children would contribute to a reduction in internalising and externalising behavior problems.

(Please see a Introduction, lines 73-75 of our revised manuscript.)

This study's hypothesis was that adherence to 24-h MG would be associated with reduced internalizing and externalizing behavioral problems in Chinese preschool children aged 3-6 years. In order to test this hypothesis, we analyzed the relationship between adherence to 24-h MG and internalizing and externalizing behavioral problems. Also, we further examined if there were differences in adherence between weekdays and weekends.

(Please see a Discussion, lines 247-252 of our revised manuscript.)
